

# Chemistry-climate model simulations of the Mt. Pinatubo eruption using CCMI and CMIP6 stratospheric aerosol data

Laura Revell[1,2], Andrea Stenke[2], Beiping Luo[2], Stefanie Kremser[3], Eugene Rozanov[2,4],
Timofei Sukhodolov[4], and Thomas Peter[2]

[1]Bodeker Scientific, Christchurch, New Zealand
[2]Institute for Atmospheric and Climate Science, ETH Zurich, Zurich, Switzerland
[3]Bodeker Scientific, Alexandra, New Zealand
[4]Physical-Meteorological Observatory/World Radiation Center, Davos, Switzerland

*Correspondence to:* Laura Revell (laura@bodekerscientific.com)

**Abstract.** To simulate the impacts of volcanic eruptions on the stratosphere, chemistry-climate models that do not include an online aerosol module require temporally and spatially resolved aerosol size parameters for heterogeneous chemistry and aerosol radiative properties as a function of wavelength. For phase 1 of the Chemistry-Climate Model Initiative (CCMI-1) and, later, for phase 6 of the Coupled Model Intercomparison Project (CMIP6) two such stratospheric aerosol data sets were

compiled, whose functional capability and representativeness are compared here. For CCMI-1, the "SAGE-4$\lambda$" data set was compiled, which hinges on the measurements at four wavelengths of the SAGE (Stratospheric Aerosol and Gas Experiment) II satellite instrument and uses ground-based Lidar measurements for gap-filling immediately after the Mt. Pinatubo eruption, when the stratosphere was optically opaque for SAGE II. For CMIP6, the new "SAGE-3$\lambda$" data set was compiled, which excludes the least reliable SAGE II wavelength and uses CLAES (Cryogenic Limb Array Etalon Spectrometer) measurements

on UARS, the Upper Atmosphere Research Satellite, for gap-filling following the Mt. Pinatubo eruption instead of ground-based Lidars. Here, we performed SOCOLv3 (Solar Climate Ozone Links version 3) chemistry-climate model simulations of the recent past (1986–2005) to investigate the impact of the Mt. Pinatubo eruption in 1991 on stratospheric temperature and ozone and how this response differs depending on which aerosol data set is applied. The use of SAGE-4$\lambda$ results in heating and ozone loss being overestimated in the lower stratosphere compared to observations in the post-eruption period

by approximately 3 K and 0.2 ppmv, respectively. However, less heating occurs in the model simulations based on SAGE-3$\lambda$, because the improved gap-filling procedures after the eruption lead to less aerosol loading in the tropical lower stratosphere. As a result, simulated temperature anomalies in the model simulations based on SAGE-3$\lambda$ for CMIP6 are in excellent agreement with MERRA and ERA-Interim reanalyses in the post-eruption period. Less heating in the simulations with SAGE-3$\lambda$ means that the rate of tropical upwelling does not strengthen as much as it does in the simulations with SAGE-4$\lambda$, which limits

dynamical uplift of ozone and therefore provides more time for ozone to accumulate in tropical mid-stratospheric air. Ozone loss following the Mt. Pinatubo eruption is overestimated by 0.1 ppmv in the model simulations based on SAGE-3$\lambda$, which is a better agreement with observations than in the simulations based on SAGE-4$\lambda$. Overall, the CMIP6 stratospheric aerosol data set, SAGE-3$\lambda$, allows SOCOLv3 to more accurately simulate the post-Pinatubo eruption period.



# 1 Introduction

The stratospheric aerosol layer is a key component of the climate system as it directly affects how incoming solar radiation is scattered in Earth's atmosphere and therefore affects the solar energy input to the climate system. While this scattering leads to cooling at Earth's surface, a warming occurs in the lower stratosphere because the aerosol particles absorb outgoing terres-

trial radiation (Boucher et al., 2013). In addition to radiative effects, stratospheric aerosol particles also provide the surface for heterogeneous chemical reactions that alter the chemical composition of the stratosphere. Heterogeneous chemical reactions convert active, ozone-destroying nitrogen oxides ($NO_x=NO+NO_2$) to nitric acid ($HNO_3$), which is a relatively long-lived reservoir for nitrogen species, therefore gas-phase $NO_x$-induced ozone destruction slows. However, with anthropogenically enhanced background concentrations of stratospheric chlorine, the efficiency of the catalytic chlorine cycles increases and

overcompensates the slowing of the $NO_x$ cycle due to a reduction of chlorine deactivation into the reservoir species $ClONO_2$ (Kinnison et al., 1994; Tie and Brasseur, 1995; Pitari et al., 2014; Muthers et al., 2015). The efficiency of the catalytic chlorine cycles is further increased by chlorine activation on aerosols under very cold conditions (polar regions or close to the tropical tropopause), i.e. heterogeneous reactions between chlorine reservoir species (e.g. HCl and $ClONO_2$) produce active, ozone-destroying chlorine radicals. Furthermore, sulfate aerosols are the basis for polar stratospheric cloud formation, and thus

influence ozone chemistry during the polar night and in spring (Carslaw et al., 1997). The resultant decreases in ozone then also affect climate at Earth's surface.

Major volcanic eruptions such as the eruption of Mt. Pinatubo in the Philippines in June 1991 inject large amounts of sulfur dioxide ($SO_2$) into the stratosphere, where $SO_2$ is oxidised to form sulfuric acid ($H_2SO_4$). Because of the extremely low vapour pressure of $H_2SO_4$-$H_2O$ solutions at lower stratospheric temperatures, gaseous $H_2SO_4$ quickly condenses forming sulfate

aerosol particles. The Mt. Pinatubo eruption is frequently studied owing to good observational coverage at that point in time. Two to four months after the eruption, the lower tropical stratosphere warmed by up to 3.5 K (Labitzke and McCormick, 1992), and global-mean surface temperatures decreased by approximately 0.4 K (Robock and Mao, 1995). Column ozone decreased by 5–10% over the globe, with large losses observed at northern midlatitudes and in the tropics (Randel et al., 1995). Small increases in ozone were initially observed at southern midlatitudes due to the aerosol-induced increase in tropical upwelling

and subsequent enhanced transport of ozone by Southern Hemisphere extratropical downwelling (Aquila et al., 2013).

Because aerosols play such an important role in Earth's climate, as observed from past volcanic eruptions (Guillet et al., 2017; Solomon et al., 2011), it is vitally important that their effects on the atmosphere can be accurately simulated. Satellite-based measurements of stratospheric aerosol properties have been available since 1979, and from these global, spatially resolved values for aerosol surface area density (SAD), mass and mean radius can be derived, albeit only with large uncertainties

because the aerosol size distribution is under-determined by the extinction or backscatter information. These quantities are needed to drive global circulation and chemistry-climate models (GCMs and CCMs), which are used for simulating aerosol effects on climate. However, model simulations of the Mt. Pinatubo eruption are diverse. For example, the CCMs participating in CCMVal-2, the predecessor activity to phase 1 of the Chemistry-Climate Model Initiative (CCMI-1), simulated global-mean temperature anomalies between -1 K and +9 K at 50 hPa in the post-Pinatubo eruption period, and global-mean ozone anoma-





lies between -2% and -22% (Mancini et al., 2010). While this is partly due to how the models handle aerosol radiative and chemical processes, the process of compiling the best historic stratospheric aerosol data set with which to drive the models is also incomplete. Kremser et al. (2016) discuss the challenges inherent in constructing a long-term stratospheric aerosol climatology, a process which is further complicated by the fact that following a major volcanic eruption, the lower stratosphere is too

optically thick for occultation instruments onboard satellites to make accurate measurements, so that a gap-filling procedure is required for the opaque regions.

Here we evaluate stratospheric aerosol data sets produced for two major modelling activities, CCMI-1 and phase 6 of the Coupled Model Intercomparison Project (CMIP6). For the ongoing CCMI-1 activity, CCMs are evaluated to obtain projections of the stratospheric ozone layer, tropospheric composition, air quality, global climate change and the interactions between them

(Morgenstern et al., 2017). For CMIP6, participating GCMs and CCMs provide projections of how the Earth system responds to forcing, and how climate may change in future (Eyring et al., 2016). Here we analyse CCM simulations forced with SAGE-$4\lambda$ and SAGE-$3\lambda$, the stratospheric aerosol data sets that were developed for CCMI-1 and CMIP6, respectively. We investigate the impact of the Mt. Pinatubo eruption on climate and stratospheric chemistry as simulated with the two stratospheric aerosol data sets. The results indicate that stratospheric temperatures and ozone changes induced by the Mt. Pinatubo eruption in

simulations based on SAGE-$3\lambda$ for CMIP6 are in better agreement with observations than the simulations based on SAGE-$4\lambda$ for CCMI-1, which in turn is better than previous aerosol data sets based on SAGE retrieval versions 5.9 or earlier (Arfeuille et al., 2013).

## 2  Stratospheric aerosol data sets

To simulate the effects of stratospheric aerosol on chemistry and climate, GCMs and CCMs need temporally and spatially

resolved values of aerosol radiative properties as a function of wavelength spanning the electromagnetic spectrum from the ultraviolet to the infrared, and information about the aerosol size distribution such as SAD and mean radii. The stratospheric aerosol data sets developed for CCMI-1 and CMIP6 are summarized in Table 1, and described in more detail below.

### 2.1  SAGE-$4\lambda$: CCMI stratospheric aerosol data set

The CCMI stratospheric aerosol data set (Luo, 2013) was prepared for models participating in CCMI-1 (Morgenstern et al.,

2017). This data set covers the period 1960–2010, and is based on SAGE (Stratospheric Aerosol and Gas Experiment) II extinction data. It constructs single mode lognormal aerosol size distributions by means of the SAGE-$4\lambda$ algorithm (Section 2.3), which uses all four wavelengths (385, 452, 525 and 1024 nm) of SAGE II data when available. In times before and after SAGE II (October 1984–August 2005) other satellite measurements are used. Since satellite measurements became available only in 1979, for volcanically quiescent periods prior to 1979, the monthly mean background aerosol measured by SAGE II during the

volcanic quiescent period 1996–2005 is used. Volcanic contributions are calculated using the AER-2D model for spatial and temporal evolution (Arfeuille et al., 2014). Stratospheric aerosol optical depths are calibrated using photometer data when available. From 1979 onwards, the data set is based on satellite measurements from the SAM, SAGE I, and SAGE II instruments,





followed by CALIPSO after 2005, which was when SAGE II measurements ended. Ground-based Lidar measurements supplement SAGE II data following the Mt. Pinatubo eruption, when much of the lower stratosphere was too opaque for SAGE II to measure (Arfeuille et al., 2013). The SAGE-4$\lambda$ data set and description can be found at ftp://iacftp.ethz.ch/pub_read/luo/ccmi/.

## 2.2 SAGE-3$\lambda$: CMIP6 stratospheric aerosol data set

The CMIP6 stratospheric aerosol data set was prepared for models participating in CMIP6 (Eyring et al., 2016), including the ongoing Model Intercomparison Project on the climatic response to Volcanic forcing (VolMIP) (Zanchettin et al., 2016). In the CMIP6 stratospheric aerosol data set, which spans from 1850–2015, SAD and radiative properties are derived via the SAGE-3$\lambda$ algorithm similar to SAGE-4$\lambda$, but omitting the less reliable channel at 385 nm (see below). Similar to the CCMI data set, for volcanically quiescent periods prior to 1979, the monthly mean background aerosol measured by SAGE II during

the volcanic quiescent period 1996–2005 is used, and volcanic contributions are calculated using the AER-2D model for spatial and temporal evolution (Arfeuille et al., 2014), calibrated by photometer data when available. From 1979 onwards, the data set is based on satellite measurements from the SAM, SAGE I, SAGE II, CALIPSO and OSIRIS instruments. CLAES (Cryogenic Limb Array Etalon Spectrometer) measurements are used for data-filling following the Mt. Pinatubo eruption. Furthermore, one additional correction is applied in SAGE-3$\lambda$ that is missing in SAGE-4$\lambda$: in the extra-tropical lowermost stratosphere

tiny particles ($r < 10$ nm), which hardly scatter or absorb sunlight and are practically invisible to satellites, but contribute significantly to SAD, are corrected based on optical particle counter (OPC) data (Deshler et al., 2003). This correction is applied below 20 km to SAD, volume density, mean radius and $H_2SO_4$ mass. No correction is made for the radiative properties, as the tiny particles hardly influence the radiative balance. (We note that we apply the original OPC data and do not use the new derivations of size distributions by Kovilakam and Deshler (2015), as they slightly overestimate SAD during volcanic periods).

The SAGE-3$\lambda$ data set and description can be found at ftp://iacftp.ethz.ch/pub_read/luo/CMIP6/.

## 2.3 SAGE-3,4$\lambda$ algorithms

The SAGE-4$\lambda$ and SAGE-3$\lambda$ algorithms were used in the CCMI and CMIP6 data sets, respectively. In both cases, a lognormal size distribution of stratospheric aerosol is assumed. For a single mode lognormal distribution, three parameters (number density $n$, mode radius $r$ and width $\sigma$) are required. Both the SAGE-3$\lambda$ and SAGE-4$\lambda$ algorithms consist of two steps:

1) Obtain $n$, $r$ and $\sigma$ by fitting the SAGE II extinction coefficients at four wavelengths (385, 452, 525 and 1024 nm) for the CCMI data set and three wavelengths for the CMIP6 data set. The data uncertainty at 385 nm is much higher than for the other three wavelengths, for two reasons. Firstly, the molecular extinction at shorter wavelengths is higher than at longer wavelengths. Therefore the removal of the signal of air molecules is much more difficult. Secondly, the extinction at 385 nm is also affected by gas phase absorption. Therefore, in the more recent CMIP6 data set, the extinction coefficients at 385 nm are

excluded.

2) Using the SAGE II data, a correction for effective radius ($r_{eff}$) and $\sigma$ with an extinction coefficient at wavelength 1020 nm ($k_{1020}$) is obtained. During the SAGE II time period, the remaining two parameters were calculated using the $r_{eff}$–$k_{1020}$ correlation for the CCMI data set and $\sigma$–$k_{1020}$ correlation for the CMIP6 data set. For other time periods, extinction coefficients





at only one wavelength were available (from satellite instruments or photometers). Both correlations, again obtained from the SAGE II time period, were used to calculate the remaining unknown parameter, the number density. At pre-photometer times, volcanic aerosol inputs were simulated using the AER stratospheric aerosol model (Arfeuille et al., 2014).

The radiative properties (extinction coefficients, single scattering albedos and asymmetry factors) for model wavelength bands were then calculated using Mie theory (Bohren and Huffman, 2007). For SOCOLv3 this procedure was executed for 6 wavelength bands in the shortwave range (solar radiation) and 16 in the longwave range (terrestrial radiation).

## 3    Chemistry-Climate Model and simulations

Simulations were performed with version 3 of the Solar Climate Ozone Links (SOCOLv3) CCM, which participated in CCMI-1 (Morgenstern et al., 2017). The base version of the CCM is described in detail and validated against observations by Stenke et al. (2013). Since then, SOCOLv3's tropospheric chemistry scheme was upgraded (Revell et al., 2015) and the model formulation was updated for participation in CCMI-1, which led to improvements in SOCOLv3's simulation of key stratospheric variables such as temperature and water vapour concentration (Revell et al., 2016). For this study, SOCOLv3 was run with 39 vertical levels between the Earth's surface and 80 km ($\sim$0.01 hPa) and T42 horizontal resolution, which corresponds to latitude/longitude grid spacing of $2.8° \times 2.8°$.

Aerosol radiative effects are calculated within SOCOLv3 by using pre-calculated extinction coefficients, asymmetry factors and single scattering albedos for each spectral band to derive radiative heating rates. Radiative transfer calculations are performed by the shortwave radiation scheme of Fouquart and Bonnel (1980), using three spectral bands in the UV-visible range and three bands in the near-IR (NIR) range (Cagnazzo et al., 2007). For the longwave part of the spectrum the RRTM (Rapid Radiative Transfer Model) by Mlawer et al. (1997) including 16 bands is used. In the stratosphere, aerosol effects on heterogeneous chemistry are taken into account by using SAD and mean aerosol radius provided within the CCMI and CMIP6 stratospheric aerosol data sets. In the troposphere an aerosol data set is prescribed (Anet et al., 2013) which affects radiation but not heterogeneous chemistry. SAD is set to zero in the troposphere, and aerosol radiative properties merge from the stratospheric to the tropospheric data sets through two transition layers at the WMO-defined tropopause. In the lower transition layer two-thirds of the prescribed tropospheric aerosol is used in combination with one-third of the prescribed stratospheric aerosol. In the upper transition layer these proportions are reversed, so that one-third of the prescribed tropospheric aerosol is used together with two-thirds stratospheric aerosol.

To better understand the chemical versus dynamical effects on stratospheric ozone following the eruption of Mt. Pinatubo, the ozone destruction rates of the gas-phase catalytic $NO_x$, reactive hydrogen ($HO_x$=H+OH+HO$_2$) and reactive chlorine ($Cl_x$=Cl+ClO+2$\times$Cl$_2$O$_2$+ClONO$_2$) cycles were tracked as a function of latitude, longitude, pressure and time as described in detail by Revell et al. (2012); see their Supporting Information for a list of the chemical cycles tracked.

Two ensembles of SOCOLv3 simulations were performed, a "CCMI" ensemble and a "CMIP6" ensemble. Each ensemble consists of five SOCOLv3 simulations, where for each simulation the initial $CO_2$ concentration was perturbed slightly to explore the model's internal variability. Both ensembles used boundary conditions recommended for the CCMI-1 reference



simulation REF-C1, except that the simulations performed for the "CMIP6" ensemble used the stratospheric aerosol data set that was prepared for CMIP6. The REF-C1 simulation is a free-running simulation (i.e., meteorology evolves without nudging to reanalyses) of the recent past, as described in detail by Morgenstern et al. (2017). Greenhouse gas concentrations follow observations until 2005, then follow Representative Concentration Pathway (RCP) 8.5 (Masui et al., 2011). Historical

ozone precursor emissions are used until 2000 (Lamarque et al., 2010), and then follow RCP 6.0 (Meehl et al., 2013). Sea surface temperatures and sea ice concentrations are prescribed following observations until 2003 (Rayner et al., 2003), and then follow RCP 6.0. Halogen-containing ozone-depleting substances follow the A1 scenario from WMO (2011) which includes observations until 2009.

Three of the "CCMI" simulations ran for the full REF-C1 period from 1960–2010 and have been uploaded to the CCMI
archive (ETH-PMOD, 2015) (excluding the model spin-up period from 1950–1959). The other seven simulations ran from 1986–2005 to focus on the Mt. Pinatubo eruption period, and have been uploaded to a separate online repository (Kuchar and Revell, 2017). Here we focus on the common period, i.e. 1986–2005 for each simulation, and examine anomalies following the Mt. Pinatubo eruption. Anomalies are calculated by removing the annual cycle averaged over 1986–2005. This approach does not completely isolate the volcanic signal, but allows a consistent comparison between data obtained from SOCOLv3,
observations and reanalyses.

## 4   Results and Discussion

### 4.1   Difference in aerosol mass in the CCMI and CMIP6 aerosol data sets

We first of all compare the SAGE-$4\lambda$ and SAGE-$3\lambda$ stratospheric aerosol data sets used for CCMI and CMIP6, respectively. The relative difference in $H_2SO_4$ aerosol mass is shown in Fig. 1. Differences in aerosol mass loading are due to the different
gap-filling procedures used following the Mt. Pinatubo eruption. Lidar data were used for gap-filling in the SAGE-$4\lambda$ data set, while CLAES data were predominantly used for the SAGE-$3\lambda$ data set (Lidar measurements were occasionally used but only in the tropics). Stark differences exist between aerosol loading in the two data sets in June 1991 (Fig. 1b), because in the SAGE-$3\lambda$ data set, missing data between $20°$ N–$20°$ S were gap-filled by replicating May 1991 data, thus effectively moving the timing of the Pinatubo eruption to July 1991 (Thomason et al., 2017).

Differences in aerosol loading lead to changes in stratospheric heating due to absorption of longwave radiation by aerosols. In the CMIP6 data set there is approximately twice as much aerosol loading in the tropical middle stratosphere (around 30–50 hPa) than in the CCMI data set (Fig. 1e–h). However in the tropical lower stratosphere, just above the tropical tropopause, there is approximately five times less aerosol in the CMIP6 data set than in the CCMI data set. Changes in heating where the stratosphere is coldest, i.e. the lower stratosphere, have a larger impact on heating than higher in the stratosphere where it is
warmer.





## 4.2 Ensemble simulations using the CCMI aerosol data set

Changes in stratospheric temperature, dynamics and chemistry following the Mt. Pinatubo eruption have been documented extensively (Pitari and Rizi, 1993; Kinnison et al., 1994; Randel et al., 1995; Rosenfield et al., 1997; Rozanov et al., 2002; Aquila et al., 2013; Pitari et al., 2014). Here we examine how the Mt. Pinatubo eruption is simulated with SOCOLv3 using the

SAGE-4$\lambda$ stratospheric aerosol data set for CCMI, and then compare the post-eruption changes in temperature, chemistry and dynamics with the simulations using the SAGE-3$\lambda$ stratospheric aerosol data set for CMIP6.

Ensemble-, zonal-mean anomalies in temperature, ozone concentrations and the rate of the vertical residual circulation for the CCMI ensemble are shown in Figure 2. Averaged over the six months following the Mt. Pinatubo eruption, volcanic aerosol causes heating in the lower stratosphere of approximately up to 4 K via absorption of NIR solar radiation and outgoing

longwave radiation (Fig. 2a). Heating in the tropical stratosphere drives an increase in the rate of tropical upwelling (Rosenfield et al., 1997), indicated by an increase in the residual vertical velocity w* (Fig. 2b), which is a useful proxy for the strength of the Brewer-Dobson circulation. As well as increased tropical upwelling, downwelling in the Southern Hemisphere is enhanced during the six months following the eruption, which is consistent with the findings of Aquila et al. (2013) and Dhomse et al. (2015). Stronger tropical upwelling leads to the ozone maximum shifting upwards, causing localized reductions in tropical

ozone concentrations of 0.4 ppmv ($\sim$10%) centered at 30 hPa (Fig. 2c).

Stratospheric ozone is not only influenced by changes in the rate of tropical upwelling but also by stratospheric composition changes following the eruption. Heterogeneous chemical reactions on the surface of aerosol particles convert active $NO_x$ to reservoir species, and reservoir chlorine to active $Cl_x$ (i.e. $HCl + ClONO_2 \rightarrow Cl_2 + HNO_3$, followed by photolysis of $Cl_2$ to produce Cl radicals). With increased conversion of $NO_x$ to reservoir species, the gas-phase ozone destroying $NO_x$ cycles slow,

less ClO is converted to $ClONO_2$, and thus the $Cl_x$ cycles become faster. This can be seen in Fig. 2d, which shows anomalies in the tropical zonal-mean rates of the gas-phase $NO_x$, $HO_x$ and $Cl_x$ ozone destruction cycles. Between 10–50 hPa, $NO_x$-induced ozone destruction slows by up to 40% in the 6 months following the eruption, indicating increased $NO_x$ deactivation. This results in a faster rate of $HO_x$- and $Cl_x$-induced ozone loss, as shown previously by e.g. Rodriguez et al. (1991) and Tie and Brasseur (1995). $HO_x$ chemistry is also faster because the tropical cold-point tropopause warms following the eruption and the

flux of water vapour into the stratosphere increases (Löffler et al., 2016). At 30 hPa the net chemical effect is reduced ozone destruction due to $NO_x$ deactivation, as shown by the black trace in Fig. 2d. However the overall effect on tropical ozone at 30 hPa is driven by the dynamical effect, i.e. a localized reduction due to uplift of ozone caused by increased tropical upwelling. The chlorine activation effect can be seen just above 100 hPa in Fig. 2d, where increased abundances of $Cl_x$ lead to the rate of the $Cl_x$-induced ozone loss cycles accelerating by up to 150%. Although this is a large relative increase, the absolute increase

is small as the gas-phase chlorine cycles are generally slow in the lowermost stratosphere (not shown).

## 4.3 Comparison of simulations using the CCMI and CMIP6 aerosol data sets

Having shown in general terms how SOCOLv3 simulates the Mt. Pinatubo eruption, we now examine the specific differences induced by the CCMI and CMIP6 stratospheric aerosol data sets. Ensemble mean differences between temperature, w*, ozone





and the rate of ozone destruction chemistry in the CCMI and CMIP6 simulations are shown in Fig. 3. While the stratosphere warms by up to 4 K in the CCMI ensemble (Fig. 2a), the tropical warming is only ∼2 K in the CMIP6 ensemble (Fig. 3a), due to less aerosol mass loading in the tropical lower stratosphere (Fig. 1). Concurrently, the tropical vertical residual circulation strengthens less in the CMIP6 ensemble compared with the CCMI ensemble (Fig. 3b), which leads to less dynamical uplift of

ozone-rich air, and a smaller local ozone reduction in the middle stratosphere is simulated (Fig. 3c and 4b).

We also examine the difference in the $NO_x$ and $Cl_x$ chemical ozone destruction rates following the eruption between the CCMI and CMIP6 ensembles (Fig. 3d). It is expected that differences in aerosol mass and SAD between the CMIP6 and CCMI stratospheric aerosol data sets lead to differences in the rate of heterogeneous chemical reactions. This can be seen clearly in the rate of the $Cl_x$ cycles in the tropical lower stratosphere. Because there is less aerosol mass in the tropical lower stratosphere

in the CMIP6 data set compared with the CCMI data set, there is less chlorine activation, and therefore $Cl_x$-induced ozone destruction is significantly slower. In the middle stratosphere more $NO_x$ deactivation, and therefore less $NO_x$-induced ozone destruction, occurs in the CCMI ensemble compared with the CMIP6 ensemble.

Finally, tropical temperature and ozone anomalies as simulated with SOCOLv3 are compared with reanalyses and observations (Fig. 4). We focus on the 30 hPa level as the preceding figures indicate that this is the pressure level around which

significant differences between ozone and temperature in the two ensembles are centered. Compared to MERRA and ERA-Interim reanalysis data, simulations using the CCMI aerosol data set overestimate the temperature response to the Mt. Pinatubo eruption by ∼3 K, which was also shown by Arfeuille et al. (2013) and Kuchar et al. (2017). The CMIP6 ensemble simulates tropical warming of ∼2 K following the eruption, which is in good agreement with ERA-Interim (Dee et al., 2011) and MERRA (Rienecker et al., 2011) reanalyses (Fig. 4a), and also with CCM simulations using a new database of volcanic $SO_2$ emissions

and plume altitudes (Mills et al., 2016).

The CMIP6 ensemble simulations of ozone in the post-eruption period show a better agreement with observations than the simulations based on the CCMI data set (Fig. 4b). Merged satellite observations from the Stratospheric Water and Ozone Satellite Homogenized (SWOOSH) data set (Davis et al., 2016) show ozone decreasing by ∼0.4 ppmv six months after the eruption. The CCMI ensemble overestimates ozone loss in this period by up to 0.2 ppmv, while the CMIP6 ensemble agrees

more closely with observations, and overestimates ozone loss only by ∼0.1 ppmv.

## 5   Conclusions

We have used two stratospheric aerosol data sets developed for the model intercomparison activities CCMI-1 and CMIP6 to drive SOCOLv3 CCM simulations of the Mt. Pinatubo eruption. Following the eruption, aerosol mass injected into the lower stratosphere absorbs infrared radiation and heats the stratosphere. This in turn leads to strengthened tropical upwelling and lifts

the ozone maximum in the tropics, causing a localized reduction in ozone around 30 hPa. In the simulations using the SAGE-4$\lambda$ stratospheric aerosol data set developed for CCMI, stratospheric warming is overestimated by approximately 3 K compared to reanalyses, and local reductions in ozone are overestimated by approximately 0.2 ppmv. Because of different gap-filling procedures used for the lower stratosphere following the Mt. Pinatubo eruption, the SAGE-3$\lambda$ data set developed for CMIP6



contains less aerosol mass in the lower stratosphere compared with the CCMI data set. Therefore in the model simulations using CMIP6 stratospheric aerosol, the stratosphere warms less than it does in the simulations with CCMI stratospheric aerosol. Using CMIP6 stratospheric aerosol, SOCOLv3 simulates a warming of approximately 2 K following the Mt. Pinatubo eruption that is in good agreement with reanalyses. While ozone loss is overestimated compared to observations, it is in closer agreement

5    with observations than the results from the simulations using CCMI stratospheric aerosol.

The CCM simulations presented here indicate that using the SAGE-3$\lambda$ stratospheric aerosol data set developed for CMIP6 in SOCOLv3 leads to an improved simulation of stratospheric temperature and ozone changes following the Mt. Pinatubo eruption compared with the simulations using the SAGE-4$\lambda$ stratospheric aerosol data set developed for CCMI. However, various CCMs and GCMs calculate the radiative and chemical effects of aerosols on the stratosphere in different ways, and the

10   two stratospheric aerosol data sets should therefore be used within other models to validate our conclusions.

*Data availability.* SOCOL v.3 CCMI-1 REF-C1 data are held at the British Atmospheric Data Centre, see http://catalogue.ceda.ac.uk/uuid/1005d2c25d14483aa66a5f4a7f50fcf0. All other simulations can be found at https://data.mendeley.com/datasets/khrhbw6wn5/draft?a=35729334-ca37-4434-bd26-60dadfb9c22b.

*Author contributions.* BL and TP were involved in developing the SAGE-4$\lambda$ and SAGE-3$\lambda$ data sets. AS, ER and TS were involved in

15   developing the SOCOLv3 CCM. AS devised the experiments. LER performed the model simulations and analysed the data. LER led the writing of this manuscript, assisted by SK and all other co-authors.

*Competing interests.* The authors declare no competing interests.

*Acknowledgements.* LER thanks Larry Thomason for helpful discussions around this paper. ER and TS acknowledge support from the Swiss National Science Foundation under grant 200021_169241 (VEC).



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



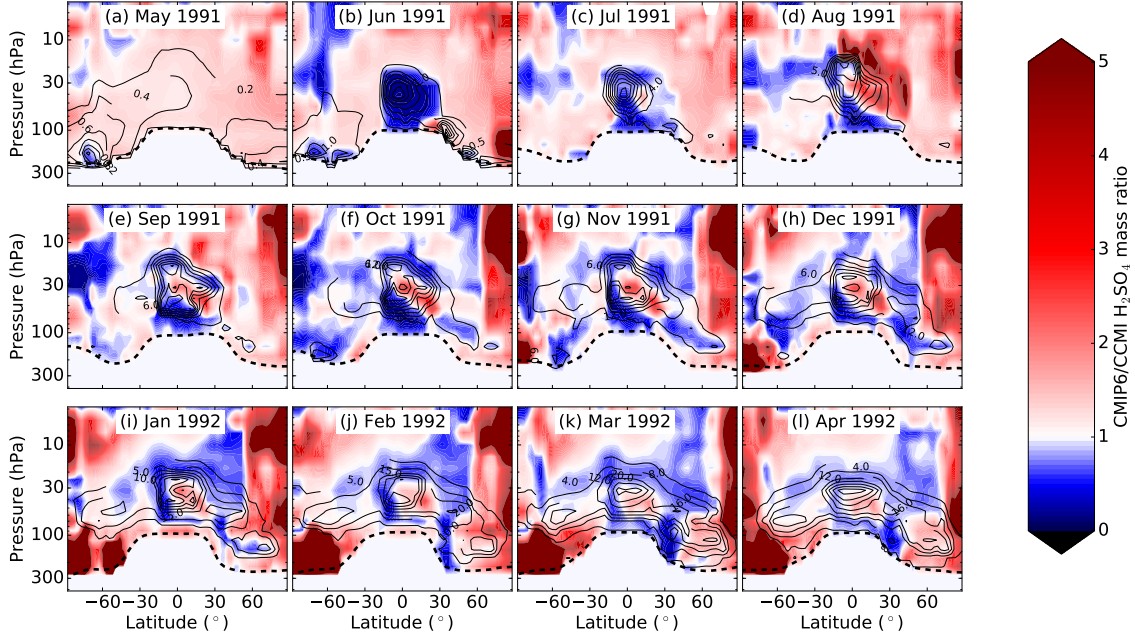

**Figure 1.** Ratio of $H_2SO_4$ aerosol mass in the CCMI and CMIP6 stratospheric aerosol data sets for 12 months around the Mt. Pinatubo eruption in June 1991, CMIP6/CCMI. Black contours show the CCMI $H_2SO_4$ mass in $10^9$ molecules cm$^{-3}$. The dashed black line shows the location of the WMO-defined tropopause. Note that aerosol data are supplied down to 5 km altitude in the CCMI and CMIP6 data sets, which are based on cloud-cleared SAGE II measurements, but these values are recommended for use only above the model tropopause.

**Table 1.** Stratospheric aerosol data sets used for CCMI-1 and CMIP6.

|  | SAGE-4$\lambda$ data set for CCMs in CCMI-1 | SAGE-3$\lambda$ data set for GCMs and CCMs in CMIP6 |
|---|---|---|
| Period | 1960-2010 | 1850-2015 |
| Data used | SAM, SAGE I, SAGE II, CALIPSO; sun-photometer data; AER stratospheric aerosol model. | SAM, SAGE I, SAGE II, SAM, CALIPSO, OSIRIS; sun-photometer data; AER stratospheric aerosol model; mass, volume density, SAD, $r_{\text{eff}}$ corrected for very small particles below 20 km by OPC measurements. |
| Data gap filling following the Mt. Pinatubo eruption | Lidar measurements | Predominantly CLAES observations |


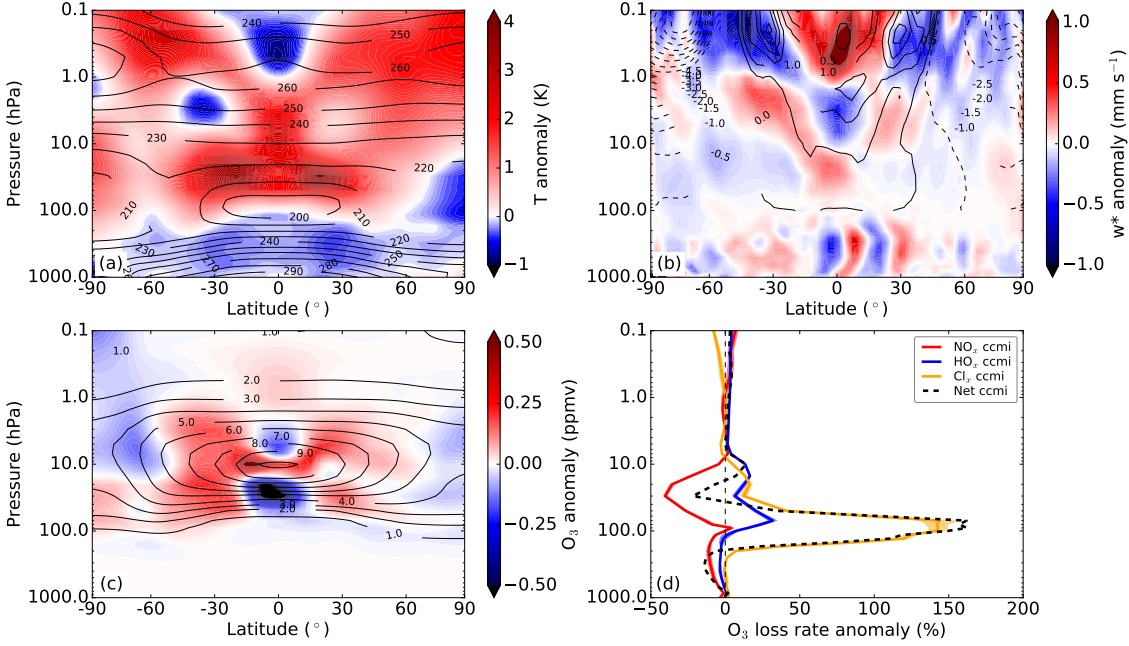

**Figure 2.** Ensemble-mean, zonal-mean anomalies averaged over the 6 months following the Mt. Pinatubo eruption for the simulations using CCMI aerosol. (a) Temperature anomalies in the CCMI ensemble (red/blue shading). Black contours indicate the annual climatological mean (1986-2005) temperatures for the CCMI ensemble. (b) As for (a), but showing anomalies in the rate of the vertical residual circulation (w*). For clarity, the annual climatological mean rate of the vertical residual circulation (black contours) is only shown above 100 hPa. Dashed contours indicate negative values. (c) As for (a), but showing ozone anomalies. (d) Anomalies in tropical ($15°$N $-15°$ S) ozone destruction rates by the $NO_x$, $HO_x$ and $Cl_x$ ozone destruction cycles. Negative anomalies indicate slower ozone destruction. Shading indicates the ensemble standard deviation. The dashed black line shows the sum of the $NO_x$, $HO_x$ and $Cl_x$ anomalies, i.e. the net anomaly in the ozone destruction rate following the eruption.





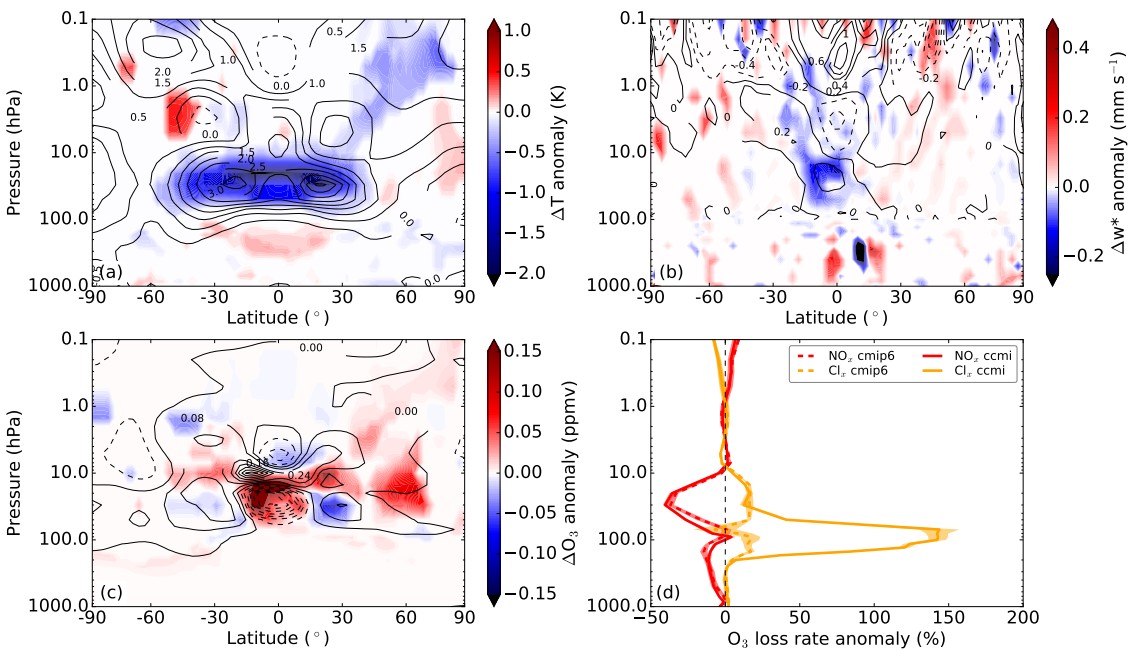

**Figure 3.** Differences in ensemble-mean, zonal-mean anomalies averaged over the 6 months following the Mt. Pinatubo eruption between the ensembles using CMIP6 and CCMI aerosol (CMIP6 minus CCMI). For the contour plots (a-c), regions that are not statistically significant at the 95% level of confidence (as calculated with Student's t test, $p < 0.05$) are set to zero. (a) Difference in temperature anomalies (red/blue shading). For reference, the black contours represent the CCMI anomalies over the same period, i.e., the red/blue shading from Fig. 2a. (b) As for (a), but showing anomalies in the vertical residual circulation. (c) As for (a), but showing ozone anomalies. (d) As for Fig. 2d, but showing anomalies in the tropical ozone destruction rates for only the $NO_x$ and $Cl_x$ ozone destruction cycles in the CMIP6 and CCMI ensembles.




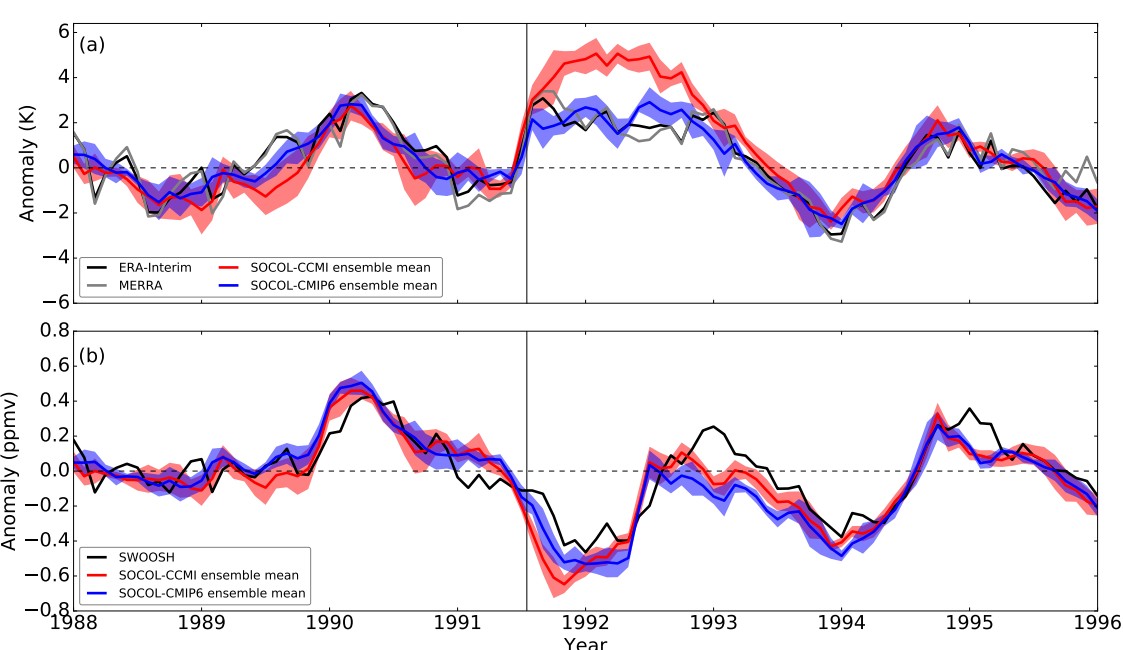

**Figure 4.** Time series of (a) temperature and (b) ozone anomalies at 30 hPa, 15° N–15° S. The red and blue lines denote the ensemble mean of the SOCOLv3 CCMI and CMIP6 ensembles, respectively. The shaded areas denote the ensemble mean plus or minus one standard deviation of simulated anomalies, and the vertical lines show the timing of the Mt. Pinatubo eruption on 15 June 1991.