# Peer review of "Impacts of Mt. Pinatubo volcanic aerosol on the tropical stratosphere in chemistry-climate model simulations using CCMI and CMIP6 stratospheric aerosol data"

_Atmospheric Chemistry and Physics, 2017_

## Referee Comment (RC1) · M. Toohey (Referee) · 1 Aug 2017

This manuscript describes coupled chemistry climate model simulations utilizing two different volcanic forcing data sets, and explores differences in the simulated responses to the volcanic forcing, including stratospheric temperatures, circulation and ozone. The experimental set-up is clear, and the results show that the stratospheric temperature anomalies produced by the newer CMIP6 volcanic forcing reconstruction are in closer agreement to observations. The paper also provides information regarding the construction of the CMIP6 forcing data, which is presently not available elsewhere.

I find the work to be well within the scope of ACP, and the conclusions to be in general

well justified by the results shown. I have a few minor comments I encourage the authors to consider before publication.

General comments

1. The results of the study focus almost exclusively on the tropics. Extratropical ozone changes in the SH are mentioned in passing, but there is no analysis of extratropical NH ozone changes (which seem to be positive for both forcing sets, inconsistent with observations), or polar temperatures (despite large changes in the forcing at high latitudes), etc. However, the title is very general, and some statements throughout the manuscript could be construed as applying to the stratosphere as a whole, rather than just the tropics (see specific comments below). I suggest the title be changed to reflect the concentration on the tropics, and some care be taken to be clear about the specificity of the results.

2. The introduction mentions the wide range simulated temperature and ozone responses to volcanic forcing in the CCMVal activity. But since the CMIP6 volcanic forcing data set is nominally an update to forcings used in past CMIP activities, it would make sense to briefly review the simulated responses to volcanic forcing in past CMIPs. Charlton-Perez et al. (2013) show CMIP5 global mean stratospheric temperature changes associated with Pinatubo split into high-and low-top models, and Driscoll et al. (2012) show tropical temperature anomalies, split into groups of models using different forcing reconstructions. Toohey et al. (2014) compare temperature and circulation anomalies from the Stenchikov forcing (basically equivalent to the Sato et al. (1993) forcing used in many CMIP5 models) and the CCMI (SAGE-4$\lambda$) forcing.

3. In a few places, the authors draw a direct line of causation from heating of the tropical lower stratosphere and increased tropical upwelling, and in some cases, link this further with increases in extratropical downwelling. While this may be true, there is also evidence of post-volcanic changes in extratropical large-scale wave breaking in observations (Graf et al., 2007; Poberaj et al., 2011) and model results (Bittner et

al., 2016; Toohey et al., 2014). This increased wave breaking should increase transport from the tropics to extratropics, and induce residual circulation anomalies. I think there is still a low degree of understanding on how the related processes of enhanced wave-breaking and tropical heating affect stratospheric tropical upwelling, extratropical mixing and downwelling. The issue of causation is not central to this study, so I'm not necessarily suggesting a detailed review of the topic, but I encourage the authors to not oversell the understanding of the mechanistic explanation of circulation changes, for example in the Introduction (p2, l24), results (p7, ll10-13) and Conclusions (p8, l29).

4. Pedantic semantic comment: these are not really simulations of the Mt. Pinatubo eruption (as stated in the title and throughout the document), they are rather simulations of the atmospheric response to stratospheric aerosols resulting from the Mt. Pinatubo eruption. This is of course obvious to many readers, but can be confusing to readers new to the field.

Specific comments

P1, l9: suggest "uses measurements from CLAES (. . .) on UARS, the. . ."

P1, l14: this overestimated heating and ozone loss is specifically in the lower tropical stratosphere. Comparisons of extratropical temperatures and ozone are not shown.

P1, l17: Again, this applies only to tropical temperatures.

P1, l20: This ozone loss is specific to 30 hPa, 15S-15N I believe, and is a peak value I guess?

P2, l2: I think the IR absorption by aerosols was known about, and reasonably well understood before 2013.

P2, l15: The last sentence here is a strong statement, which could use some support from prior work. Son et al. (2010) comes to mind, but there are surely other references that would support this.

[Figure]

P2, l26: This statement is supported by two pretty random references from a great sea of literature. At the very least, an "e.g." is called for, otherwise review articles (e.g., Robock, 2000; Timmreck, 2012) would seem to be a better fit.

P3, l13: I don't think the analysis of tropical lower stratospheric temperatures really constitutes an investigation of "climate".

P4, l13-20: Some more details regarding the correction are needed. It is written that the issue pertains to the "extra-tropical lowermost stratosphere", but later the correction is applied "below 20 km", is this at all latitudes or only in the extratropics? Does the correction increase or decrease the SAD, by roughly how much, and is it seasonally varying? It is written the correction applies to H2SO4 mass, should this not affect IR absorption then, which is roughly proportional to aerosol mass?

P4, l31-: This got confusing for me. If $n$, $r$ and $\sigma$ are found in step 1, what are the "remaining two parameters" mentioned in step 2? And with some constructed relationships between $k_{1020}$ and $r_{eff}$ and $\sigma$, how can you use these to calculate number density (p5, l2)? I really wonder how one can retrieve 3 pieces of information ($n$, $r$ and $\sigma$) from measurements at a single wavelength, there must be some assumptions that go into this reconstruction.

P5, l16: "...derive heating rates." But also scattering, atmospheric transmission, etc.

P6, l2: The simulations are free-running, but what about the QBO? The temperature anomalies in Fig 4 seem to be oscillating quasi biennially, with the simulations right in line with the observations.

P6, l28-30: This last sentence seems at least misplaced (in this subsection on the aerosol mass comparison), and also not well supported by any results shown here.

P8, l14: Focusing on a single height level always runs the risk of sampling error. Observed temperature anomalies after Pinatubo appear to peak around 20 hPa, slightly higher than most simulations (see Fig 1 of Toohey et al., 2014). Just to be sure that

30 hPa is telling the right (and/or full) story, it would really be great to include a latitude/height cross section of zonal mean temperature anomalies (and perhaps ozone too).

P8, l31: the temperature and ozone anomalies quoted here are specific to locations and times.

P9, l7: suggest "tropical stratospheric temperature..."

Fig 1: It's hard to read anything quantitative from the color scale used in this plot. Perhaps percent difference (CMIP6-CCMI)/CCMI would work better?

Fig 3: Showing the results of the CMIP6 simulations as a difference plot wrt the CCMI forcing is very useful, but on the other hand, it would also be nice to see the results in their absolute values. Many potentially interesting results are hard to glean from only the difference plot, for example, it's clear that the upwelling in the lower tropical stratosphere is decreased in the CMIP6 simulations compared to CCMI, but it's not obvious then what the magnitude of the upwelling anomaly is in the CMIP6 simulation ensemble. Such a plot could (rather easily I assume) be added to the main text or included as a supplement.

References

Bittner, M., Timmreck, C., Schmidt, H., Toohey, M. and Krüger, K.: The impact of wave-mean flow interaction on the Northern Hemisphere polar vortex after tropical volcanic eruptions, J. Geophys. Res. Atmos., doi:10.1002/2015JD024603, 2016.

Charlton-Perez, A. J., Baldwin, M. P., Birner, T., Black, R. X., Butler, A. H., Calvo, N., Davis, N. A., Gerber, E. P., Gillett, N., Hardiman, S., Kim, J., Krüger, K., Lee, Y.-Y., Manzini, E., McDaniel, B. A., Polvani, L., Reichler, T., Shaw, T. A., Sigmond, M., Son, S.-W., Toohey, M., Wilcox, L., Yoden, S., Christiansen, B., Lott, F., Shindell, D., Yukimoto, S. and Watanabe, S.: On the lack of stratospheric dynamical variability in low-top versions of the CMIP5 models, J. Geophys. Res. Atmos., 118(6), 2494–2505,

doi:10.1002/jgrd.50125, 2013.

Driscoll, S., Bozzo, A., Gray, L. J., Robock, A. and Stenchikov, G.: Coupled Model Intercomparison Project 5 (CMIP5) simulations of climate following volcanic eruptions, J. Geophys. Res., 117(D17), doi:10.1029/2012JD017607, 2012.

Graf, H.-F., Li, Q. and Giorgetta, M. A.: Volcanic effects on climate: revisiting the mechanisms, Atmos. Chem. Phys., 7(17), 4503–4511, doi:10.5194/acp-7-4503-2007, 2007. Poberaj, C. S., Staehelin, J. and Brunner, D.: Missing Stratospheric Ozone Decrease at Southern Hemisphere Middle Latitudes after Mt. Pinatubo: A Dynamical Perspective, J. Atmos. Sci., 68(9), 1922–1945, doi:10.1175/JAS-D-10-05004.1, 2011.

Robock, A.: Volcanic Eruptions and Climate, Rev. Geophys., 38(2), 191–219, doi:10.1029/1998RG000054, 2000.

Sato, M., Hansen, J. E., McCormick, M. P. and Pollack, J. B.: Stratospheric Aerosol Optical Depths, 1850–1990, J. Geophys. Res., 98(D12), 22987–22994, doi:10.1029/93JD02553, 1993. Timmreck, C.: Modeling the climatic effects of large explosive volcanic eruptions, Wiley Interdiscip. Rev. Clim. Chang., 3(6), 545–564, doi:10.1002/wcc.192, 2012.

Son, S.-W., Gerber, E. P., Perlwitz, J., Polvani, L. M., Gillett, N. P., Seo, K.-H., Eyring, V., Shepherd, T. G., Waugh, D., Akiyoshi, H., Austin, J., Baumgaertner, A., Bekki, S., Braesicke, P., Brühl, C., Butchart, N., Chipperfield, M. P., Cugnet, D., Dameris, M., Dhomse, S., Frith, S., Garny, H., Garcia, R., Hardiman, S. C., Jöckel, P., Lamarque, J. F., Mancini, E., Marchand, M., Michou, M., Nakamura, T., Morgenstern, O., Pitari, G., Plummer, D. A., Pyle, J., Rozanov, E., Scinocca, J. F., Shibata, K., Smale, D., Teyssèdre, H., Tian, W. and Yamashita, Y.: Impact of stratospheric ozone on Southern Hemisphere circulation change: A multimodel assessment, J. Geophys. Res., 115(7), D00M07, doi:10.1029/2010JD014271, 2010.

Toohey, M., Krüger, K., Bittner, M., Timmreck, C. and Schmidt, H.: The impact of volcanic aerosol on the Northern Hemisphere stratospheric polar vortex: mechanisms and sensitivity to forcing structure, Atmos. Chem. Phys., 14(23), 13063–13079, doi:10.5194/acp-14-13063-2014, 2014.
* * *
**[ACPD](ACPD)**

---

## Referee Comment (RC2) · Anonymous Referee #2 · 3 Aug 2017

In this manuscript the authors compare the results of SOCOLv3 simulations performed using the SAGE-4\lambda and SAGE-3\lambda stratospheric aerosol datasets, used for the CCMI-1 and CMIP-6 model intercomparisons, respectively. In particular, the authors compare the temperatures and ozone concentrations during the post-Pinatubo period in the two simulation ensembles to each other and to the MERRA and ERA-Interim reanalysis.

I have found this an interesting paper, well-written and logically organized. It is a good paper that represent a necessary reference to document the differences between the two datasets. I have only some minor comments:

[Figure]

- page 3 line 13: "we investigate the impact of the Mt. Pinatubo eruption on climate and stratospheric chemistry". The authors only show changes in temperature and w*, too little to speak about changes in climate. I would explicitly write "we investigate the impact of the Mt. Pinatubo eruption on stratospheric temperatures and chemistry".

- section 2.3 is not very clear. Starting from the title, I would spell out the full names of the databases: "The SAGE-3$\lambda$ and SAGE-4$\lambda$". Initially I wondered if the authors where introducing a third database that merges SAGE-3$\lambda$ and SAGE-4$\lambda$. Secondarily, I do not understand the steps. Step 1 is the calculation of n, r, and $\sigma$ from the different wavelengths. But what is step 2? Which correction is calculated? Or did you mean "correlation"? Also, what are the remaining two parameters, n and $\sigma$? But they have already been obtained in step 1.

- figure 4: The authors compare with MERRA and ERA-Interim to establish which one of the two databases lead to better simulations. However, reanalysis might not be the best tool to evaluate a model after a volcanic eruption, as they are driven by satellite data which might not be reliable after such strong perturbation. Additionally, they might not respond correctly to such a strong and sudden perturbation. I would suggest to add a comparison to measurements, many of which as cited in the introduction.
* * *

---

## Author Comment (AC1) · 27 Sep 2017

*M. Toohey (Referee)*

*This manuscript describes coupled chemistry climate model simulations utilizing two different volcanic forcing data sets, and explores differences in the simulated responses to the volcanic forcing, including stratospheric temperatures, circulation and ozone. The experimental set-up is clear, and the results show that the stratospheric temperature anomalies produced by the newer CMIP6 volcanic forcing reconstruction are in closer agreement to observations. The paper also provides information regarding the construction of the CMIP6 forcing data, which is presently not available elsewhere. I find the work to be well within the scope of ACP, and the conclusions to be in general well justified by the results shown. I have a few minor comments I encourage the authors to consider before publication.*

*General comments*

*1. The results of the study focus almost exclusively on the tropics. Extratropical ozone changes in the SH are mentioned in passing, but there is no analysis of extratropical NH ozone changes (which seem to be positive for both forcing sets, inconsistent with observations), or polar temperatures (despite large changes in the forcing at high latitudes), etc. However, the title is very general, and some statements throughout the manuscript could be construed as applying to the stratosphere as a whole, rather than just the tropics (see specific comments below). I suggest the title be changed to reflect the concentration on the tropics, and some care be taken to be clear about the specificity of the results.*

**We have changed the title to: "Impacts of Mt. Pinatubo volcanic aerosol on the tropical stratosphere in chemistry-climate model simulations using CCMI and CMIP6 stratospheric aerosol data."**

*2. The introduction mentions the wide range simulated temperature and ozone responses to volcanic forcing in the CCMVal activity. But since the CMIP6 volcanic forcing data set is nominally an update to forcings used in past CMIP activities, it would make sense to briefly review the simulated responses to volcanic forcing in past CMIPs. Charlton-Perez et al. (2013) show CMIP5 global mean stratospheric temperature changes associated with Pinatubo split into high-and low-top models, and Driscoll et al. (2012) show tropical temperature anomalies, split into groups of models using different forcing reconstructions. Toohey et al. (2014) compare temperature and circulation anomalies from the Stenchikov forcing (basically equivalent to the Sato et al. (1993) forcing used in many CMIP5 models) and the CCMI (SAGE-4) forcing.*

Thank you for bringing these papers to our attention; we thought that Driscoll et al. (2012) was most relevant for the study at hand and we have included the range of CMIP5 tropical temperature anomalies in the introduction:

"However, model simulations of the Mt. Pinatubo eruption are diverse. For example, GCMs participating in phase 5 of the Coupled Model Intercomparison Project (CMIP5) simulated tropical (30 °N–30 °S) temperature anomalies at 50 hPa ranging between 2–10 K (Driscoll et al., 2012). Similarly, CCMs participating in CCMVal-2, the predecessor activity to phase 1 of the Chemistry-Climate Model Initiative (CCMI-1), simulated global-mean temperature anomalies between -1 K and +9 K at 50 hPa in the post-Pinatubo eruption period, and global-mean ozone anomalies between -2% and -22% (Mancini et al., 2010)."

*3. In a few places, the authors draw a direct line of causation from heating of the tropical lower stratosphere and increased tropical upwelling, and in some cases, link this further with increases in extratropical downwelling. While this may be true, there is also evidence of post-volcanic changes in extratropical large-scale wave breaking in observations (Graf et al., 2007; Poberaj et al., 2011) and model results (Bittner et al., 2016; Toohey et al., 2014). This increased wave breaking should increase transport from the tropics to extratropics, and induce residual circulation anomalies. I think there is still a low degree of understanding on how the related processes of enhanced wave-breaking and tropical heating affect stratospheric tropical upwelling, extratropical mixing and downwelling. The issue of causation is not central to this study, so I'm not necessarily suggesting a detailed review of the topic, but I encourage the authors to not oversell the understanding of the mechanistic explanation of circulation changes, for example in the Introduction (p2, l24), results (p7, ll10-13) and Conclusions (p8, l29).*

We have taken care to moderate our discussion on the links between heating and upwelling, e.g.:

"Heating in the tropical stratosphere is thought to drive an increase in the rate of tropical upwelling (Rosenfield et al. 1997), indicated by an increase in the residual vertical velocity w* (Fig. 2b), which is a useful proxy for the strength of the Brewer-Dobson circulation."

*4. Pedantic semantic comment: these are not really simulations of the Mt. Pinatubo eruption (as stated in the title and throughout the document), they are rather simulations of the atmospheric response to stratospheric aerosols resulting from the Mt. Pinatubo eruption. This is of course obvious to many readers, but can be confusing to readers new to the field.*

Good point and we hope that the new title, and various changes throughout the manuscript, will be less confusing to readers new to the field.

*Specific comments*

*P1, l9: suggest "uses measurements from CLAES (: : :) on UARS, the: : :"*

**Changed as suggested:**

**"For CMIP6, the new "SAGE-3λ" data set was compiled, which excludes the least reliable SAGE II wavelength and uses measurements from CLAES (Cryogenic Limb Array Etalon Spectrometer) on UARS, the Upper Atmosphere Research Satellite, for gap-filling following the Mt. Pinatubo eruption instead of ground-based Lidars."**

*P1, l14: this overestimated heating and ozone loss is specifically in the lower tropical stratosphere. Comparisons of extratropical temperatures and ozone are not shown.*

**We have clarified that the overestimations apply only to the tropical lower stratosphere:**

**"The use of SAGE-4λ results in heating and ozone loss being overestimated in the tropical lower stratosphere compared to observations in the post-eruption period by approximately 3 K and 0.2 ppmv, respectively."**

*P1, l17: Again, this applies only to tropical temperatures.*

**We have clarified this in the text:**

**"As a result, simulated tropical temperature anomalies in the model simulations based on SAGE-3λ for CMIP6 are in excellent agreement with MERRA and ERA-Interim reanalyses in the post-eruption period."**

*P1, l20: This ozone loss is specific to 30 hPa, 15°S-15°N I believe, and is a peak value I guess?*

**Yes, we have now noted that the ozone loss is overestimated by up to 0.1 ppmv:**

**"Ozone loss following the Mt. Pinatubo eruption is overestimated by up to 0.1 ppmv in the model simulations based on SAGE-3λ, which is a better agreement with observations than in the simulations based on SAGE-4λ."**

*P2, l2: I think the IR absorption by aerosols was known about, and reasonably well understood before 2013.*

**True, although the IPCC assessments provide a good summary of the current state of knowledge for readers new to the field. We have amended this to "see e.g. Boucher et al. (2013) and references therein."**

*P2, l15: The last sentence here is a strong statement, which could use some support from prior work. Son et al. (2010) comes to mind, but there are surely other references that would support this.*

**We have cited Son et al. (2010) here.**

*P2, l26: This statement is supported by two pretty random references from a great sea of literature. At the very least, an "e.g." is called for, otherwise review articles (e.g., Robock, 2000; Timmreck, 2012) would seem to be a better fit.*

**Changed as suggested:**

**"Because aerosols play such an important role in Earth's climate, as observed from past volcanic eruptions e.g. (Robock, 2000; Solomon et al., 2011; Guillet et al., 2017)..."**

*P3, l13: I don't think the analysis of tropical lower stratospheric temperatures really constitutes an investigation of "climate".*

**Changed to "stratospheric chemistry and temperatures."**

*P4, l13-20: Some more details regarding the correction are needed. It is written that the issue pertains to the "extra-tropical lowermost stratosphere", but later the correction is applied "below 20 km", is this at all latitudes or only in the extratropics? It is written the correction applies to H2SO4 mass, should this not affect IR absorption then, which is roughly proportional to aerosol mass?*

**The correction is an altitude-dependent value (shown in Figure 1 below), applied everywhere irrespective of season or latitude, and increases SAD by the amount shown in Figure 1. These small particles (i.e. the SAD included by the correction) are included in the model's chemistry scheme but are not considered in the radiative calculations. We took only the contribution of the log-normal distribution for both longwave and shortwave. The comparison of extinctions at 3.46 µm shows good agreement with HALOE data in the stratosphere.**

[Figure]

SAD correction

h [km] (y-axis)
Delta SAD [um2/cm3] (x-axis)

**Figure 1: the SAD OPC correction applied to SAGE-3λ data.**

*P4, l31-: This got confusing for me. If n, r and σ are found in step 1, what are the "remaining two parameters" mentioned in step 2? And with some constructed relationships between $k_{1020}$ and $r_{eff}$ and σ, how can you use these to calculate number density (p5, l2)? I really wonder how one can retrieve 3 pieces of information (n, r and σ) from measurements at a single wavelength, there must be some assumptions that go into this reconstruction.*

In step 1 we obtained three parameters (n, r and σ) using only three input values (extinction coefficients at three wavelengths for CMIP6 aerosols/4 wavelengths for CCMI aerosols), which are partially correlated. A small measurement error of input values may cause large excursions of the output parameters (n, r, σ). Therefore, we use a σ -$k_{1020}$ correlation in CMIP6 to minimize the effects introduced by the measurement errors, even during the SAGE II time, where extinction coefficients at 3 wavelengths were available. This correlation is obtained from the output of step 1.

For some time periods, extinction coefficients at only one wavelength were available (from satellite instruments or photometers). Both correlations (σ - $k_{1020}$ and $r_{eff}$–$k_{1020}$), again obtained from the SAGE II time period, were used to calculate the remaining unknown parameter, the number density.

We have rewritten Section 2.3 as:

"The SAGE-4λ and SAGE-3λ algorithms were used in the CCMI and CMIP6 data sets, respectively. In both cases, a lognormal size distribution of stratospheric aerosol is assumed. For a single mode lognormal distribution, three parameters (number density n, mode radius r and width σ) are required. Both the SAGE-3λ and SAGE-4λ algorithms consist of two steps for calculating n, r and σ:

1) Obtain n, r and σ by fitting the SAGE II extinction coefficients at four wavelengths (385, 452, 525 and 1024 nm) for the CCMI data set and three wavelengths for the CMIP6 data set. The data

uncertainty at 385 nm is much higher than for the other three wavelengths, for two reasons. Firstly, the molecular extinction at shorter wavelengths is higher than at longer wavelengths. Therefore the removal of the signal of air molecules is much more difficult. Secondly, the extinction at 385 nm is also affected by gas phase absorption. Therefore, in the more recent CMIP6 data set, the extinction coefficients at 385 nm are excluded.

2) In step 1, n, r and σ were obtained using the extinction coefficients at three wavelengths for the CMIP6 data set, and four wavelengths for the CCMI data set, which are partially correlated. However, a small measurement error on the input values may cause large inaccuracies in the output parameters (n, r and σ). Therefore a σ –k1020 correlation was used in the CMIP6 data set to minimize the effects introduced by the measurement errors, even during the SAGE II period, where extinction coefficients at three wavelengths were available. This correlation is obtained from the output of step 1. In CCMI, the $r_{eff}$–$k_{1020}$ correlation was used to obtain r. The remaining two parameters (n and σ) were obtained by fitting to the measured extinction coefficients. The fitting quality remains almost as good as step 1.

For other time periods (outside the SAGE II period), extinction coefficients at only one wavelength were available (from satellite instruments or photometers). Both correlations (σ –$k_{1020}$ and $r_{eff}$–$k_{1020}$), again obtained from the SAGE II time period, were used to calculate the remaining unknown parameter, the number density. The radiative properties (extinction coefficients, single scattering albedos and asymmetry factors) for model wavelength bands were then calculated using Mie theory (Bohren and Huffman, 2007). For SOCOLv3 this procedure was executed for 6 wavelength bands in the shortwave range (solar radiation) and 16 in the longwave range (terrestrial radiation)."

*P5, l16: ": : :derive heating rates." But also scattering, atmospheric transmission, etc.*

**Changed as suggested:**

"Aerosol radiative effects are calculated within SOCOLv3 by using pre-calculated extinction coefficients, asymmetry factors and single scattering albedos for each spectral band to derive radiative heating rates, scattering and atmospheric transmission."

*P6, l2: The simulations are free-running, but what about the QBO? The temperature anomalies in Fig 4 seem to be oscillating quasi biennially, with the simulations right in line with the observations.*

**Yes, the QBO is nudged to observations. We added:**

"Although the REF-C1 simulation is free-running, in SOCOLv3 the quasi-biennial oscillation (QBO) is forced by nudging the zonal tropical stratospheric winds (20° N—20° S, 3-90 hPa) to a time series of observed wind profiles (Giorgetta et al. 1996, Stenke et al. 2013)."

*P6, l28-30: This last sentence seems at least misplaced (in this subsection on the aerosol mass comparison), and also not well supported by any results shown here.*

**We have removed this sentence.**

*P8, l14: Focusing on a single height level always runs the risk of sampling error. Observed temperature anomalies after Pinatubo appear to peak around 20 hPa, slightly higher than most simulations (see Fig 1 of Toohey et al., 2014). Just to be sure that 30 hPa is telling the right (and/or full) story, it would really be great to include a latitude/height cross section of zonal mean temperature anomalies (and perhaps ozone too).*

**Zonal-mean temperature and ozone anomalies from reanalyses/observations are shown as a function of pressure and latitude in Figure 2 below (panels a-c), and we will include these in a supplementary figure to the paper. These show that 30 hPa is an appropriate pressure level to focus on for our time series plots, with both reanalyses showing peak temperature anomalies at 20-30 hPa (a-b) and SWOOSH observations showing maximum ozone decreases at 30 hPa (c).**

[Figure]

**Figure 2: Top row: Anomalies in the 6 months following the Mt. Pinatubo eruption for (a) MERRA temperature reanalyses; (b) ERA-Interim temperature reanalyses; (c) SWOOSH ozone observations. Black contour lines show the annual climatological mean (1986-2005). Bottom row: Ensemble-mean, zonal-mean anomalies averaged over the 6 months following the Mt. Pinatubo eruption for the simulations using CMIP6 aerosol. (a) Temperature anomalies in the CMIP6 ensemble (red/blue shading). Black contours indicate the annual climatological mean (1986-2005) temperatures for the**

**CMIP6 ensemble. (b) As for (a), but showing anomalies in the rate of the vertical residual circulation (w\*). For clarity, the annual climatological mean rate of the vertical residual circulation (black contours) is only shown above 100 hPa. Dashed contours indicate negative values. (c) As for (a), but showing ozone anomalies.**

*P8, l31: the temperature and ozone anomalies quoted here are specific to locations and times.*

**We have clarified that these refer to tropical stratospheric anomalies following the Mt. Pinatubo eruption:**

**"In the simulations using the SAGE-4λ stratospheric aerosol data set developed for CCMI, tropical stratospheric warming following the Mt. Pinatubo eruption is overestimated by approximately 3 K compared to reanalyses, and local reductions in ozone are overestimated by approximately 0.2 ppmv."**

*P9, l7: suggest "tropical stratospheric temperature: : :"*

**Changed as suggested.**

*Fig 1: It's hard to read anything quantitative from the color scale used in this plot. Perhaps percent difference (CMIP6-CCMI)/CCMI would work better?*

**This figure has been updated as suggested:**

[Figure]

**Figure 3: Percent difference in $H_2SO_4$ aerosol mass in the CCMI and CMIP6 stratospheric aerosol data sets for 12 months around the Mt. Pinatubo eruption in June 1991, CMIP6 minus CCMI. Black contours show the CCMI H2SO4 mass in $10^9$ molecules $cm^{-3}$. The dashed black line shows the location of the WMO-defined tropopause. Note that aerosol data are supplied down to 5 km altitude in the CCMI and CMIP6 data sets, which are based on cloud-cleared SAGE II measurements, but these values are recommended for use only above the model tropopause.**

*Fig 3: Showing the results of the CMIP6 simulations as a difference plot wrt the CCMI forcing is very useful, but on the other hand, it would also be nice to see the results in their absolute values. Many potentially interesting results are hard to glean from only the difference plot, for example, it's clear that the upwelling in the lower tropical stratosphere is decreased in the CMIP6 simulations compared to CCMI, but it's not obvious then what the magnitude of the upwelling anomaly is in the CMIP6 simulation ensemble. Such a plot could (rather easily I assume) be added to the main text or included as a supplement.*

**We have prepared a figure showing the absolute values for the CMIP6 simulations to include in the supplement; see Figure 2 above, panels d-f.**

---

## Author Comment (AC2) · 27 Sep 2017

*In this manuscript the authors compare the results of SOCOLv3 simulations performed using the SAGE-4λ and SAGE-3λ stratospheric aerosol datasets, used for the CCMI-1 and CMIP-6 model intercomparisons, respectively. In particular, the authors compare the temperatures and ozone concentrations during the post-Pinatubo period in the two simulation ensembles to each other and to the MERRA and ERA-Interim reanalysis. I have found this an interesting paper, well-written and logically organized. It is a good paper that represent a necessary reference to document the differences between the two datasets. I have only some minor comments:*

*- page 3 line 13: "we investigate the impact of the Mt. Pinatubo eruption on climate and stratospheric chemistry". The authors only show changes in temperature and w\*, too little to speak about changes in climate. I would explicitly write "we investigate the impact of the Mt. Pinatubo eruption on stratospheric temperatures and chemistry".*

**Changed as suggested.**

*- section 2.3 is not very clear. Starting from the title, I would spell out the full names of the databases: "The SAGE-3λ and SAGE-4λ". Initially I wondered if the authors where introducing a third database that merges SAGE-3λ and SAGE-4λ. Secondarily, I do not understand the steps. Step 1 is the calculation of n, r, and σ from the different wavelengths. But what is step 2? Which correction is calculated? Or did you mean "correlation"? Also, what are the remaining two parameters, n and σ? But they have already been obtained in step 1.*

**We have changed the title of this section as suggested. And yes, "correlation" was meant rather than "correction" – thanks for bringing this error to our attention! Step 2 describes how n and σ can be obtained from the correlation even when SAGE II data are not available. We have rewritten step 2 as:**

**"In step 1, $n$, $r$ and σ were obtained using the extinction coefficients at three wavelengths for the CMIP6 data set, and four wavelengths for the CCMI data set, which are partially correlated. However, a small measurement error on the input values may cause large inaccuracies in the output parameters ($n$, $r$ and σ). Therefore a σ -$k_{1020}$ correlation was used in the CMIP6 data set to minimize the effects introduced by the measurement errors, even during the SAGE II period, where extinction coefficients at three wavelengths were available. This correlation is obtained from the output of step 1. In CCMI,**

the $r_{eff}$-$k_{1020}$ correlation was used to obtain $r$. The remaining two parameters ($n$ and $\sigma$) were obtained by fitting to the measured extinction coefficients. The fitting quality remains almost as good as step 1.

For other time periods (outside the SAGE II period), extinction coefficients at only one wavelength were available (from satellite instruments or photometers). Both correlations ($\sigma$ -$k_{1020}$ and $r_{eff}$-$k_{1020}$), again obtained from the SAGE II time period, were used to calculate the remaining unknown parameter, the number density."

*- figure 4: The authors compare with MERRA and ERA-Interim to establish which one of the two databases lead to better simulations. However, reanalysis might not be the best tool to evaluate a model after a volcanic eruption, as they are driven by satellite data which might not be reliable after such strong perturbation. Additionally, they might not respond correctly to such a strong and sudden perturbation. I would suggest to add a comparison to measurements, many of which as cited in the introduction.*

We note that the reanalyses assimilate all available data, not just satellite data. Further, Dee et al. (2011) note that in ERA-Interim they apply a bias correction which avoids some of the problems encountered in the post-Pinatubo eruption period in the ERA-40 reanalysis. Zonal-mean latitude/pressure cross-sections of temperature anomalies in the MERRA and ERA-Interim reanalyses (Figure 1 below) show warming in the tropical lower stratosphere of ~3 K, which, given that this is a 6-monthly average, is in good agreement with the "up to 3.5 K" warming reported by Labitzke and McCormick, 1992 (cited in the introduction).

[Figure]

Figure 1: Anomalies in the 6 months following the Mt. Pinatubo eruption for (a) MERRA temperature reanalyses; (b) ERA-Interim temperature reanalyses; (c) SWOOSH ozone observations. Black contour lines show the annual climatological mean (1986-2005).